# CSR Statements in International and Czech Luxury Fashion Industry at the Onset and during the COVID-19 Pandemic—Slowing Down the Fast Fashion Business?

**Radka MacGregor Pelikánová \***, **Tereza Němečková** and **Robert K. MacGregor**

Department of International Business, Metropolitan University Prague, 100 00 Prague, Czech Republic; tereza.nemeckova@mup.cz (T.N.); robertkmacgregor@yahoo.com (R.K.M.)
\* Correspondence: radkamacgregor@yahoo.com; Tel.: +420-725-555-312

**Abstract:** COVID-19 leads to a reinforced need to address sustainability at the economic, environmental, and social pillars, and the six categories of Corporate Social Responsibility (CSR) of businesses, that is, their self-commitment to integrate sustainability in their policies and strategies and to become more effective and efficient. Luxury fashion businesses refer heavily but not identically, to their CSR, by formally issuing CSR reports and Codes of Ethics, and informally voicing their pro-CSR statements. Then, the COVID-19 pandemic hit and brought important changes. This burning issue translates into three objectives in this paper—assessing the CSR statements of International and Czech Luxury Fashion Industry businesses (i) at the onset of the COVID-19 pandemic, (ii) during the COVID-19 pandemic, and (iii) identifying new trends. To address them, a holistic and interdisciplinary case study exploration was performed entailing 10 International Luxury Fashion businesses and 10 Czech Luxury Fashion businesses. The resulting data were explored via Meta-Analysis and content analysis, teleological interpretations, etc. The critical and comparative review revealed six interesting trends indicating a shift in perception of sustainability and CSR caused by COVID-19. The review offers propositions for how the COVID-19 threats could become an opportunity for rebuilding this industry.

**Keywords:** corporate social responsibility (CSR); COVID-19; luxury fashion industry; sustainability

## 1. Introduction

Our post-modern global society wants to enjoy ongoing prosperity and so organically expresses the desire to have the capacity for constant existence, that is, sustainability. The roots of sustainability go back to the concept of *Nachhaltigkeit*, which was presented in 1713 by the Sachsen top mining administrator, Hans Carl von Carlowitz, in his influential book *Sylvicultura Oeconomica*. Hans Carl von Carlowitz was the pupil of the famous French minister of finance, Jean Baptiste Colbert, who brought the French economy back from bankruptcy by supporting manufacturing, equal taxing, inventors' protections, etc. Interestingly, *Nachhaltigkeit* was directly linked to the long-term responsibility vis-à-vis the environment and available resources, in particular, the forest and wood industry. Even more interestingly, in 1832, Emil André published an influential book, *Einfachste den höchsten Ertrag und die Nachhaltigkeit ganz sicher stellende Forstwirthschafts-Methode*, in Prague, in which he stressed the importance of long-term responsibility in dealing with resources. Ultimately, the *Nachhaltigkeit* became linked to the universal perpetuitas, that is, the move from the long-term to the eternal responsibility and the move from thinking regionally to globally was completed in the 20th century [1]. The merely economic and tangible resource concerns were complemented by intangible concerns and a focus on human rights by the United Nations (UN), with its Universal Declaration of Human Rights (UDHR) in 1948, which significantly contributed to the emergence of the modern concept of sustainability. The idea of sustainability is based on environmental, social, and economic pillars while focusing on the reconciliation of available resources as an increasing world

population emerged [2]. It represents a value judgment about the reconciliation of the needs of the current generation and the ability of future generations to meet their needs, thereby unifying the economic (profit), environmental (planet), and social (people) dimensions of continuity [3]. In the 1960s, the trend was magnified by the reinforced interest in socially progressive values, along with political awareness, under the auspices of *"communitarism"*. In the 1970s, this was transformed into an individualist focus due to a myriad of world and regional crises and a move from Keynesian economic theory to neoliberal theory [4]. The UN closely followed this evolution and brought a set of important global sustainability documents. The two most important of these are (i) the Report of the World Commission on Environment and Development Report: Our Common Future prepared by the Brundtland Commission [5], published as the UN Annex to document A/42/427 in 1987 ("Brundtland Report 1987") and (ii) the Resolution made during a historic UN Summit in September 2015, entitled "Transforming our world: The 2030 Agenda for Sustainable development ("UN Agenda 2030")", which brought with it its 17 Sustainable Development Goals (SDGs) and 169 associated targets [6,7].

The materialization of the concept of sustainability is feasible only with universal support, that is, a multi-stakeholder model and cross-sector partnership are desirable [8], if not necessary [9]. Therefore, the initial focus on the sustainability of states was progressively paralleled by corporate responsibility concerns of businesses and even individuals [10]. Indeed, sustainability in the narrow sense (with rather systematic and visionary features and designed for soft law and self-regulation) and corporate responsibility (with rather normative and moral features, designed for national law regulation) have converged into the modern Corporate Social Responsibility (CSR) to be assumed by businesses [11–13]. Since the concept of sustainability and CSR are critical at the global, regional, and even national levels, they were incorporated in EU strategic priorities for 2010–2020, that is, in the Europe 2020: A strategy for smart, sustainable and inclusive growth (Europe 2020) [14], in particular in the Green Paper: Promoting a European Framework for CSR [15]. Indeed, the EU repeatedly expressed its commitment to the UN Agenda, its determination to reach all 17 SDGs by 2030, and its desire to even engage businesses, investors, and consumers (see e.g., the updated Directive 2013/34/EU) [4]. The EU expects CSR to be a transparent dialogue and interaction between businesses and other stakeholders. It should be materialized by an active embracement, development of potentials [16], and proud [15] and detailed informing about it [17]. There are studies about the (lack of) CSR and CSR reporting progress by automotive [10], food [18], tobacco [19,20], agriculture, [21] and other industries [22] in central Europe [23,24], as well as comparable settings [25].

The fashion industry is arguably the world's third-largest manufacturing industry after automotive and technology industries [26]. Over the last two decades, the global fashion business shifted towards a business model that offers (the perception of) clothes at affordable prices. Such a "fast-fashion" model assumes a highly responsive supply chain that can deliver a product assortment that is periodically changing [27]. However, such a model has led to devastating social and environmental impacts. Today, the fashion industry generates huge waste and pressure on the environment [26] and is allegedly the second largest industrial polluter in the world. This is because many chemicals used in this field of business are harmful to the environment as well as to factory workers and consumers. Additionally, these businesses increased material production to such an extent that the current fast-fashion practices result in large amounts of textile waste [28]. Luxury fashion, which is the focus of this article, is on the contrary, based on uniqueness, value, and exclusivity [29]. Thus, it is often perceived as the opposite of the "fast-fashion" model marked by low prices, fast product rotations, and one that encourages over-consumption [26]. However, this perception has been widely distorted by the collapse of the Rana Plaza factory in 2013, killing more than 1000 people. The Rana Factory products were, among others, destined to go to Dior and Saint Laurent [30]. Plus, various scandals in firms, such as Prada being accused of exploiting illegal Chinese immigrants, or Louis Vuitton allowing cruelty (torturing crocodiles) as well as being removed from the FTSE4Good Index [29],

measuring the performance of companies demonstrating strong environmental, social, and governance practices, have come to the fore. These indicate that even this fashion industry segment is being exposed to serious challenges relating to sustainability and thus raising concerns about its "slow model".

The year 2020 might be another milestone as the COVID-19 pandemic "slowed down" the fashion business to the extent that many companies are struggling to stay afloat and to address supply and demand disruptions along with supply chain changes [31]. One of the most recent symptoms of the most likely deep changes coming in the near future is the open letter sent by Giorgio Armani in April 2020 to the prestigious American fashion journal, Women's Wear Daily, calling for a slower fashion movement and announcing plans to realign collections with seasons in store [32]. Indeed, COVID-19 is a challenge for many humans' certainty, but at the same time, it is a great opportunity to create a real sustainable renaissance [33]. Nevertheless, this requires the engagement of as many stakeholders as possible, that is, a multi-stakeholder model and cross-sector partnership [8,9].

The European fashion industry has been consistently playing-up sustainability concerns, including SDGs, and proclaiming CSR engagement [34]. This is even more true now than ever before when the demand for sustainable products is increasing, especially among the Millennials, the core consumers of fast-fashion products [29]. It synergistically attempts to develop luxury brands protected as trademarks accordingly [35]. Indeed, luxury brands are pillars of luxury fashion marketing and oscillate around the concepts of scarcity, exclusivity, and limited resources. Since the luxury fashion industry is about fine quality work with limited resources along with particular services [36], the marketing command to go for CSR is organically parallel with the internal operation calling for CSR [37]. In sum, CSR and CSR reporting are truly embedded in the luxury fashion industry in the EU, and customers have come to expect it [38] as much as investors [39]. At least until 2020, theoretically all luxury fashion businesses in the EU were expected to vigorously target at least some of the six CSR categories and proudly report about it [17]. However, what was the reality? What happened to it due to COVID-19? Can we detect patterns and, if yes, about what do they testify? Do we have a twilight or dawning? Are COVID-19 and its aftermaths forcing luxury fashion businesses out of the market or are they more about an opportunity to go for a genuine CSR renaissance? Is it about slowing down the wasteful fast-fashion business, or is it not? There is a need to perform a deep micro-case study about the CSR proclamations, declarations, and other statements by luxury fashion businesses in one EU member state while addressing three objectives assessing the CSR statements of International and Czech Luxury Fashion Industry businesses (i) at the onset of the COVID-19 pandemic, (ii) during the COVID-19 pandemic, and (iii) identifying new trends. In order to address these three objectives, after this introduction (1), a theoretical background is presented, including a literature and legislative review (2). Then follow the employed material and methods linked to the case study (3), with the results and discussion addressing the timing dynamics of CSR statements (4), and aftermath observations (5). The culmination of the critical exploration and performed argumentation leads to the conclusions (6).

## 2. Theoretical Background

Both the concept of sustainability and CSR are outcomes of the belief that businesses, especially large global companies, as powerful economic, social and political actors, should participate in the multi-stakeholder model and be pro-sustainability [40,41]. CSR means the responsibility towards all stakeholders, that is, towards the entire society, while addressing all three sustainability pillars [38] and even fair competition [42–44]. Responsibility, as such, has Latin roots, see *respondere*, and means that someone has to answer for the effects caused by him to an authority and this authority evaluates its damages [1]. CSR per se is not legally enforceable and does not represent a legal duty leading to legal liability, instead, CSR is an ethical responsibility entailing virtue ethics, utilitarian ethics, and deontological ethics [1]. CSR means moral obligations of the given business towards the

entire society, asking it to go beyond the mere concept of profit maximization [45]. Arguably, businesses are currently expected to show the institutionalization of sustainable and ethical principles and practices [22]. In central Europe, these trends need to be observed while considering the dynamics between business actions dealing with goods vs. services [36]. Further, CSR goes through a process of progression from none to an over-facultative to a mandatory regime [46]. Businesses are increasingly expected to communicate the social and environmental effects of their economic actions to various stakeholders [17].

Consequently, the EU has newly set a legal duty for large public-interest entities with more than 500 employees to include, in the management report, a non-financial statement linked to CSR [46], see Directive 2013/34/EU, Directive (EU) 2017/1132 and Regulation 2015/884 as updated [15]. The EU motto "united in diversity" neatly fits in with the perception of CSR as a dialogue and interaction between businesses, corporations, and their stakeholders [47] including customers [43] and is reflected by EU policies [48,49], which, so far, lead to only one legal duty regarding CSR for only some subjects—to report about it. Thusly, so far, it is up to the discretion of the businesses how much they will engage with CSR and its six categories, how they use it as competition leverage [44], and inform about it [10]. Some businesses may perceive the commitment to sustainability via CSR as a negative burden generating costs without return, that is, waste, while others can treat CSR as an impulse for improvement in all three spheres of sustainability (economic, environmental, and social), an instrument to improve their own financial performance [50] and a foundation for their marketing [37] and other strategies [51].

Arguably, the recognition of shared value policies and principles linked to CSR should lead to "a more sophisticated form of capitalism" [52] and the evolution should go from the CSR cultural reluctance over the CSR cultural grasp to a CSR cultural embedment [38]. This line of thought includes the stakeholder theory, pursuant to which business engagement with CSR leads to value creation, an improvement of the business reputation [53] and branding [54], and ultimately an increase in market share [25]. Several studies suggest that CSR brings benefits to all stakeholders and enhances financial performance [45]. CSR reporting should strongly impact a firm's value [55] and R&D spending, in particular, should boost productivity and ultimately lead to product differentiation and entry barriers [56]. Business ethics should influence the success and profits of modern businesses, and their role is set to increase in the future [57]. Social relations could bring many advantages to a business, such as employee stability, that is, human resource retention, improvement of local community relationships, and even the attraction of social and ethical investors and customers [58].

In contrast, traditional theories are more skeptical; point to limitations due to possible agency conflicts between managers, shareholders, environmental activists, etc. [46]; and underline that resource allocation due to CSR, especially for social goals, may add to the costs and consequently prevent profit maximization [59]. Several studies pointed out the negative impacts of CSR activities and spending by indicating that CSR practices can generate unnecessary costs, cripple financial results [60], and undermine the competitive advantage [61].

Modern businesses should properly reconcile the profitability, growth, and social relationships [50] and recognize that CSR cannot be totally avoided [62,63]. Society has become more sensitive about ethical [22], social [64], and environmental [65] issues; appropriate, reasonable, and well-oriented CSR "expenses" should be compensated, offset, by the advertising effect on an improved brand image [66], stable revenues from loyal clients, improved employee productivity [67], decreased risks [68], and reduced capital costs [69]. CSR needs to be properly tailored [6,7], cost-effective [54] and well disseminated. CSR statements reduce the information asymmetry and arguably even lead to a reduction of the cost of debt [70]. This is true in general, and, regarding the luxury fashion industry, in particular. The term "luxus" entails both the attractive beauty and element of exclusive excessiveness [71]. The fashion luxury industry businesses built upon it and underlined the heritage of a barely affordable, sophisticated show-off [38,72]. Prima

facia, this should be fully in compliance with all three pillars of sustainability—limited resources are used for lavish "eternal" products [73], that is, luxury branding should be highly pro-sustainability and luxury fashion marketing tools [74] should take advantage of all six CSR categories [10]. In addition, newly emerging fashion customer groups—the HENRYs (High-Earners-Not-Rich-Yet) and youngsters (Millennials and Gen Z) seem very CSR-aware and expect pro-CSR statements [75]. This trend should be even more magnified by the COVID-19 pandemic and related crises, events, and issues. The luxury fashion industry expects such statements in both CSR reports and Codes of Ethics, which are a form of self-regulation that contain general principles to guide behavior. Additionally, they are multifunctional, because they serve to represent and enhance a business's culture and value as well as leading to the adoption of a specific organization and/or governance structure [34]. The term "ethics" comes from the Greek (*ethikos*—customary; *ethos*—custom and/or person's fundamental orientation toward life) and leads to a theory of morality, which attempts to systematize moral judgments [57] or moral principles employed in the decision-making process. It is a vehicle to distinguish between good and bad [22]. This distinction should become even more relevant with the emergence of the COVID-19 pandemic.

Indeed, the COVID-19 pandemic, with over 4 million confirmed cases, led, among other things, to a dramatic loss of revenue and a general economic decline. Pursuant to the International Monetary Fund (IMF), the COVID-19 pandemic has brought a global economic downturn the likes of which has not been experienced since the 1870s [76]. This grim view is backed even by EU figures and the European Commission expects that the GDP of EU countries will contract by 7.5% in 2020 [76]. In reaction to that, the European Commission president, Ursula Von der Leyen, made a set of crucial statements concerning COVID-19 and its aftermath, while emphasizing that "we must not hold on to yesterday´s economy as we rebuild" [76], "we have to push for investment and reform— and we have to strengthen our economies by focusing on our common priorities, like the European Green Deal, digitalization and resilience" [77] and that it is necessary to support Europe in its transition to "a climate-neutralized and resilient economy" [78]. Consequently, conventional competition concerns [43], the drive for technological and other potentialities [16,79] along with unconventional sustainability via CSR [80], and the employment of the multi-stakeholder model [81] acquired an additional function [82,83]. That is, in short, to use the CSR drive not only to support smart, sustainable, and inclusive growth but also to make it the instrument addressing the pandemic COVID-19 as not a threat but rather as an opportunity. Perhaps it can be suggested that the prior crises made the luxury fashion industry less fun and more free AKA Fendi's fun-fur by Karl Lagerfeld [84] vs. Gucci/Prada fur-free [85] and the current COVID-19 led to even more universal, and at the same time revolutionary, changes [86]—to truly slow down the luxury fashion industry, to abolish the hectic and restless carousel of dozens and dozens of haute couture *défilés* de mode and collections annually [87] and perhaps to lead to the total victory of the fur-free material reconsideration [88]. However, all the above mentioned is merely an outcome of theoretical elaborations, academic analyses, and political proclamations. So, what is the reality? What do the CSR statements of International and Czech Luxury Fashion Industry businesses include and imply (i) at the onset of the COVID-19 pandemic and (ii) during the COVID-19 pandemic? Can we (iii) identify new trends? Is the academic proposition about the increased drive for CSR statements and its acceleration due to COVID-19 truth, or merely a chimera? Manifestly, a field search and practical study and observations need to be done in order to address these burning issues: Has the Covid-19 pandemic changed the perception of CSR activities among companies? Do they, after this pandemic experience, highlight more sustainable approaches to production and the sales process? Can we say that the COVID-19 has been a positive factor in CSR development?

## 3. Materials and Methods

The research methodology, including mined, extracted, and analyzed data and employed methods, in this paper are determined by its three mutually related objectives assessing the CSR statements of the International and Czech Luxury Fashion Industry business (i) at the onset of the COVID-19 pandemic, (ii) during the COVID-19 pandemic, and (iii) identifying new trends. Therefore, an interdisciplinary case study is performed with respect to 10 International Luxury Fashion businesses and 10 Czech Luxury Fashion businesses. This multi-disciplinary research of predominantly primary data calls for processing by both critical and comparative methods. Namely, the exploration of the yielded data employed, mainly meta-analysis and content analysis, along with a simplified Delphi method, while focusing slightly more on qualitative than quantitative aspects [89].

The case study format matches with the mentioned objectives because it allows the authors, as investigators, to retain the holistic and meaningful characteristics of real-life events [90]. The case study deals with a homogeneous sample of businesses—luxury fashion businesses. The selection of the luxury fashion businesses included was done based on the obvious quartet factor—a strong brand (reputation within the public), strong trademark (well-known trademark), strong product presentation (*défilé de mode*—fashion show), and strong product channeling (shops in the proximity of the local "5th Avenue") [72]. Consequently, and fully in compliance with the concept of luxury fashion uniqueness [14,29,38,66], only 20 fashion businesses satisfied these criteria and were included in the case study. Further, they were split into two groups—international luxury fashion businesses with foreign owners and/or headquarters and the principal operation abroad and local Czech luxury fashion businesses with local owners and/or headquarters and principal operations in the Czech Republic. The number of international luxury fashion businesses reached 10 and each and every one of these businesses are located in the Czech 5th Avenue—Pařížská street. CSR reports for 2018 are available for all of them via their domains and the national eJustice or regional BRIS portal, except for Dolce & Gabanna, while COVID-19 statements are scattered on the Internet, see Table 1. Further, these 10 businesses have a high coefficient for CSR and Environment, Social, Governance Metrics (CSR/ESG) [91].

**Table 1.** Case Study—10 International fashion businesses and their key parameters.

| Group | Business | Origin | # | Pre-COVID-19 Statements | COVID-19 Statements (Accessed on 3 March 2021) |
|---|---|---|---|---|---|
| LVMH | Louis Vuitton | 1854, Paris | 3 | CSR Rep 2018 | www.lvmh.com |
| LVMH | Christian Dior | 1946, Paris | 4 | CSR Rep 2018 | www.lvmh.com |
| LVMH | Fendi | 1925, Rome | 12 | CSR Rep 2018 | www.lvmh.com |
| LVMH | Bulgari | 1884, Rome | 13 | CSR Rep 2018 | www.lvmh.com |
| Kering | Gucci | 1921, Tuscany | 9 | CSR Rep 2018 | www.gucci.com |
| Kering | Bottega Veneta | 1966, Vicenza | 14 | CSR Rep 2018 | www.theglassmagazine.com |
| Coty | Escada | 1978, Munich | 21 | CSR Rep 2018 | - |
| Prada | Prada | 1913, Milano | 16 | CSR Rep 2018 | www.pradagroup.com |
| Tod's | Tod's | 1920, St.E.Mare | 13 | CSR Rep 2018 | - |
| D&G | Dolce & Gabanna | 1985, Milano | 28 | Code of Ethics | www.elle.com |

Source: Prepared by the Authors based on their own research of the Internet domains of businesses.

In order to achieve a comparison with local counterparts, 10 Czech businesses active in the fashion industry and perceived, at least locally, as luxury and brand-strong were selected. Particular attention was paid to their national particularity, that is, all of them were incorporated in the Czech Republic by Czechs building and/or developing traditional high-quality Czech fashion products, and all of them have always had the form of a Czech company, that is, under the Czech jurisdiction and still (at least partial) Czech ownership. Naturally, these 10 fashion businesses are less glamorous than the selected 10 international luxury fashion businesses and not included in the CSR/ESG database [91], but still, they share a similar drive for quality, exclusivity, and reputation. Table 2 summarizes them,

their line of business, origin, and the source of their CSR statements—manifestly much more dispersed than in the case of the international luxury fashion businesses.

**Table 2.** Case Study—10 Czech fashion businesses and their key parameters.

| Business | Type | Origin | Pre-COVID-19 Statements | COVID-19 Statements (Accessed on 3 March 2021) |
|---|---|---|---|---|
| Alpine Pro | Top outdoor | 1994, Brno | Code of Eth | www.alpinepro.cz/ |
| Bandi Vamos Mens | Formal | 2012, Ostrava | www.bandi.cz | www.bandi.cz |
| Blažek Praha | Formal | 1997, Praha | www.blazek.cz/ | www.blazek.cz/ |
| Evona | Underwear | 1992, Chrudim | https://www.evona.cz | www.evona.cz chrudimskenoviny.cz/ kategorie/zpravy/rousky-zdarma-usije-evona |
| Kara | Formal, leather | 1997, Trutnov | www.kara.cz | https://www.kara.cz/ |
| Moira | Underwear | 2001, Praha | Annual report | moira.cz www.seznamzpravy.cz |
| Pietro Filipi | General fashion | 1998, Praha | Annual report | www.pietro-filipi.com |
| Timo | Underwear | 1992, Praha | Annual report | www.timo.cz |
| Tonak | Hats | 1990, Nový Jičín | Annual Report | www.tonak.cz/ www.idnes.cz/ostrava/zpravy/ rousky-cena-mesta.A200325_54 0685_ostrava-zpravy_woj www.triola.cz |
| Triola | Underwear | 1994, Praha | www.triola.cz | www.seznamzpravy.cz/clanek/ stoleta-triola-jde-do-nanoprumyslu-obrat-urychlil-koronavirus-105470 |

Source: Prepared by the Authors based on their own research of the Internet domains of businesses.

In order to boost the academic robustness and practical relevancy, both official and non-official, formal and informal CSR statements of these businesses were considered. Naturally, the authors of these statements were competent persons and agents, that is, owners, CEOs, members of top management and spokespersons, and heads of communications and marketing. These statements were included in annual reports, CSR reports, Codes of Ethics and e-newsletters, and other e-campaigns, that is, they were researched on the Internet via keywords. Therefore, the principal source was the Internet, namely www pages placed on the domains of these businesses and the eJustice/BRIS portal. All processed statements were considered and categorized based on the type of business (International vs. local Czech) and the timeline with the milestone March 2020 (CSR statements pre-COVID-19 statements vs. CSR statements during COVID-19). Due to the schedule for filling financial and non-financial reports with the Czech Commercial Register, and ultimately with eJustice, basically all pre-COVID-19 CSR statements were formal CSR reports placed on eJustice, while all COVID-19 CSR statements were rather informal CSR proclamations placed on the domains of businesses.

The CSR statements, regardless of whether included in annual reports, special CSR reports, or somewhere else, along with the Codes of Ethics, have to be explored by employing the qualitative-quantitative text analysis, that is, content analysis [92]. Content analysis is a technique that makes replicable and valid inferences about texts [93] and has a long tradition in the research of CSR reporting and is considered an established research method [94]. The authors are strong proponents of the manual processing of the qualitative content analysis via the Delphi method [95] and consider automatic scanning of key words processing of the content analysis as merely auxiliary [10]. Consequently, they prefer reading over scanning, and thus they employ a simplified Delphi method ranking by three experts classifying the provided categorized CSR information (+) or (++) or (+++)

and quoting. Logically, the strongest and most concrete CSR statements got (+++) and the weakest (+) or even (0) and the six well established CSR categories were [10]:

- environment protection [65,96],
- employee matters [67,97–99],
- social matters and community concerns [64],
- respect for human rights [54],
- anti-corruption and bribery matters [22] and
- R&D activities [100].

Namely, three CSR experts, different from the authors (E.D.C., Z.F.L., L.M.), with college degrees and at least 15 years of experience in the field of law and economics, were selected, and each of them read all the entailed statements, provided comments on them and/or ranked them while considering all six CSR categories and compared their rankings. They then moved on to the next round to reduce discrepancies between the rankings by these three experts while adding direct citations from statements in order to boost the immediate authenticity and to allow refreshing glossing and Socratic questioning [101]. This holistic manual processing via the simplified Delphi method takes advantage of meta-analysis, which is an analysis of analyses [102] and which is a quasi-statistical analysis of a large collection of results from individual studies with the goal to integrate their findings [103]. The foundation of the meta-analysis is the belief that there was discovered and/or is available more than what was understood. One of the empirical options verifying the sufficient academic robustness of the performed meta-analysis is the holistic aftermath observation, that is, to see if propositions implied by the meta-analysis are at least partially projected in the new evolution and trends.

## 4. Results and Discussion

The pioneering case study entails the assessment of CSR statements of 10 international and 10 Czech Luxury Fashion Industry businesses via their CSR reports, before the COVID-19 pandemic, and via their CSR online proclamations in the Spring of 2020, that is, during the COVID-19 pandemic, with the aim to identify new trends.

### 4.1. CSR Statements at the Onset of COVID-19 Pandemic

Considering the legal framework and implied duties of businesses regarding the publication of financial and even non-financial statements annually, typically by June of the following year, the last available CSR reports of businesses are generally CSR reports for 2018 which were filed during 2019, and in 2020 they are available via the eJustice/BRIS portal.

For 2018, 9 of the 10 included luxury fashion businesses issued, via CEOs and made freely, electronically, available, reports including CSR information. The missing CSR report for Dolce & Gabanna was substituted by its Code of Ethics. These statements were manually explored via the Delphi method while paying particular attention to six CSR categories. Table 3 below, summarizes this data.

The LVHM group, that is, Louis Vuitton, Christian Dior, Fendi, and Bulgari, has been issuing annual CSR Reports, usually 50 to 100 pages long. The "LVMH 2018 Social Responsibility Report" comprises 54 pages and directly, as well as indirectly, mentions employee matters along with social, human rights, and anti-corruption matters. In contrast to these explicit and concrete statements, proclamations regarding the environment and R&D are rather general and/or underdeveloped. The International CSR dimension is obvious, the LVHM group works with UNICEF, see the "Save the Children" program, and references to partnerships and 17 SDGs are manifest.

The Kering group, that is, Gucci and Bottega Veneta, has issued a "2018 Integrated Report" of 38 pages with a clear top priority—the environment. The Kering group committed to reduce $CO_2$ emissions by 50% (Science-based target) to ensure 100% traceability in the Group's key raw materials and to achieve the highest standards in animal welfare. Concerning employee matters and social matters, community concerns, and R&D, merely general

statements were included, while a clear commitment even for suppliers was expressed with respect to respect for human rights (origin of gold or diamonds). Anti-corruption and bribery matters are not mentioned. The Code of Ethics of Kering has, similarly to LVHM, a global human rights dimension and refers to its own fundamental values as well as the UDHR and other UN documents, including the UN Global Compact and associated SDGs, various International Labor Organization conventions, and the OECD Guidelines for Multinational Enterprises along with the aim of reducing the $CO_2$ emissions by 50% in 2025 and supporting animal welfare.

**Table 3.** Case Study—10 International fashion businesses and their 2018 CSR statements.

| Group | Business | Environ | Emp. | Social | HumR | Xcorru. | R&D |
|-------|----------|---------|------|--------|------|---------|-----|
| LVMH | Louis Vuitton | + | +++ | ++ | ++ | ++ | + |
| LVMH | Christian Dior | + | +++ | ++ | ++ | ++ | + |
| LVMH | Fendi | + | +++ | ++ | ++ | ++ | + |
| LVMH | Bulgari | + | ++ | ++ | ++ | ++ | + |
| Kering | Gucci | ++ | + | + | +++ | 0 | + |
| Kering | Bottega Veneta | +++ | + | + | +++ | 0 | + |
| Coty | Escada | + | + | 0 | 0 | + | ++ |
| Prada | Prada | +++ | ++ | ++ | 0 | 0 | +++ |
| Tod's | Tod's | +++ | ++ | +++ | +++ | ++ | ++ |
| D&G | Dolce & Gabanna | + | + | + | + | + | + |

Source: Prepared by the Authors based on their own research of the Internet domains of businesses.

The Coty group with Escada stills perceives non-financial reporting as auxiliary and a mere addendum to financial reporting. Therefore the CSR report itself was reduced to one single page for 2018 and no true elaboration was proposed. Perhaps the only CSR highlight was a recognition of the R&D and Intellectual Property (IP) importance, see license presentation.

Prada, meanwhile, has prepared an "Annual Report 2018" 244 pages long, and mentions employees frequently. Nevertheless, a deep critical study reveals that the category covered in most detail is environment protection (see the Prada drive for 100% renewable resources (energy)) and its commitment to originality and independence. Back in 2007, Prada was one of the first luxury industry businesses to adopt a Code of Ethics and is also a promoter of many sustainability activities. It supports the "Manifesto of sustainability for Italian Fashion" and is a member of the "Sustainability, Ecology and Environment Commission" that aims at creating shared environmental and ethical standards. The Code of Ethics of Prada has a much less International reference character than LVHM and Kering, instead, it develops national (Italian) law compliance. Since 2019, Prada collaborates with the Fur Free Alliance and goes for "free-fur".

Tod's has issued a "2018 Annual Report" which has 277 pages and evenly pays attention to all six CSR categories. Attention is paid to ethics, tradition, creativity, and solidarity, see "Solidarity and Italian Spirit". The Code of Conduct of Tod's is similar to Prada's Code of Ethics, and this probably is due to its nature and origin, both Prada and Tod's are national Italian shareholder companies, that is, public limited companies—SpA. Similar to Prada, great stress is posited on compliance with the law. Dolce & Gabanna even issued a Code of Ethics basically covering all six CSR categories and underlying the national Italian particularism, along with general ethics concerns.

In sum, pre-COVID-19 statements of the international luxury fashion businesses are similar (even identical within the same holding group), include all six CSR categories, and pay the strongest attention either to the environment or employee categories. However, a study of pre-COVID-19 statements by the Czech luxury fashion businesses provides a less satisfactory picture, see Table 4.

**Table 4.** Case Study—10 Czech fashion businesses and their 2018 CSR statements.

| Group | Source-Comments | Environ | Emp. | Social | HumR | Xcorru. | R&D |
|-------|-----------------|---------|------|--------|------|---------|-----|
| Alpine Pro | Code of Ethics | ++ | + | ++ | ++ | + | + |
| Bandi V. | Reports, www | 0 | 0 | 0 | 0 | 0 | 0 |
| Blažek P. | Reports, www | 0 | 0 | 0 | 0 | 0 | + |
| Evona | Reports, www | 0 | 0 | 0 | 0 | 0 | 0 |
| Kara | Reports, www | 0 | 0 | 0 | 0 | 0 | 0 |
| Moira | Annual Report | ++ | ++ | + | + | + | ++ |
| Pietro Filipi | Annual Report | + | + | 0 | 0 | 0 | + |
| Timo | Annual Report | +++ | ++ | + | 0 | 0 | +++ |
| Tonak | Annual Report | + | + | 0 | 0 | 0 | + |
| Triola | www | 0 | 0 | 0 | 0 | 0 | ++ |

Source: Prepared by the Authors based on their own research of the Internet domains of businesses.

Alpine Pro has a strong orientation towards free trade, ethical concerns, and both environmental and social (community) concerns. Alpine Pro is with its CSR much more oriented outside and wants to be an ambassador of values, while internal and employee matters are not particularly mentioned. Bandi Vamos, at least until 2020, has not demonstrated any particular interest in CSR and sticks with its "modern formal macho". A direct competitor, Blažek Praha has a very similar strategy and pays little attention to CSR—the only exception is an interest in R&D in order to improve qualities of used materials (e.g., suits suitable for machine washing). Evona, with stockings and underwear, has "passed" on the possibility to go for CSR and report about it via CSR statements, Code of Ethics, online or otherwise. The same can be observed with Kara and this is highly interesting, especially considering the international move from fun fur to free fur along with massive animal welfare campaigns pushed by international luxury fashion businesses. A different picture is offered by the annual report of Moira, which deals with all six CSR categories and especially elaborates on the importance of environmental friendliness, respect for employees, and support of R&D. Although Pietro Filipi is in the same holding group with Kara, its annual report includes both financial and non-financial reporting. However, only three CSR categories are covered, and this even in a very basic format (environment, employees, and R&D) covers all six CSR categories. A similar annual report for 2018 was filed by Timo, but it must be underlined that Timo's CSR commitment is much deeper and more factual, especially regarding the environment and R&D. The 2018 annual report of Tonak includes basic information about the environment, employees and R&D. Triola's annual report for 2018 does not mention CSR, but its www mentions special R&D projects.

Pre-COVID-19 statements of the Czech luxury fashion businesses differ and the only common point is that they all lag far behind the CSR proclamations offered by pre-COVID-19 statements from the international luxury fashion businesses. Of these businesses, relatively the most CSR committed are Timo, Moira, and Alpine Pro. Directly competing businesses have not followed the same patterns and even the sub-industry choice (formal clothes, underwear, sportswear) or use of materials (fur—Kara) were irrelevant for the determination of the CSR statements and their publication. What changes, if any, did the COVID-19 pandemic bring to CSR and its reporting in the luxury fashion segment?

*4.2. CSR Statements during the COVID-19 Pandemic*

Spring 2020 was marked by the COVID-19 pandemic and all EU businesses have been impacted by it and attempted, sometimes with the help of the EU and state policies, to manage the implied challenges. Some businesses have been more resourceful, and consequently more successful, than others.

Even cursory research, based on a mere Internet browsing, reveals that the luxury fashion businesses suffered dramatically negative impacts from the COVID-19 pandemic, that their economic performance dropped, or, better to say, sunk deeply, and that they did not hesitate to look into CSR for some help to improve their reputation and perception by the public-at-large. It is argued that these CSR statements during the COVID-19 pandemic

mean a universal revolutionary change (Cápová, 2020a)—to truly slow down the luxury fashion industry, to abolish the hectic and restless carousel of dozens and dozens of haute couture défilés de mode and collections annually (Cápová, 2020b), and to lead to a total victory in re the fur free material reconsideration (Fur Free Alliance, 2020b). Table 5 shows the most important and disseminated CSR statements of the selected 10 international luxury fashion businesses via www during Spring 2020, that is, during the COVID-19 pandemic, specifically on March 3, 2021. They were researched based on the domains of these businesses and by e-research via keywords "name of business" and "COVID-19".

**Table 5.** Case Study—10 International fashion businesses and their 2020 CSR statements.

| Group | Business | Environ | Comments |
|---|---|---|---|
| LVMH<br>LVMH<br>LVMH<br>LVMH | Louis Vuitton<br>Christian Dior<br>Fendi<br>Bulgari | . . . been actively engaged since the outbreak of the Covid-19 crisis to help battle the spread of the virus . . . their spirit of solidarity through aid for medical personnel, . . . donations, . . . | Social—new production (global) |
| Kering | Gucci | Gucci will stand with its global community to fight the COVID-19 pandemic by making two separate donations of 1 million euros each to crowd funding | Social—donation (global) |
| Kering | Bottega Veneta | Funding full two-year scholarships in Italy to aid the research. support the healthcare workers €300,000... pneumology research. | Social/R&D—funding (regional) |
| Coty | Escada | - | - |
| Prada | Prada | The Prada Group is providing financial support to . . . a Hospital in Milan that will investigate why COVID-19 affects men more severely than women, | Social/R&D—funding (regional) |
| Tod´s | Tod´s | - | - |
| D&G | Dolce & Gabanna | Dolce & GabannaFund COVID-19 | Social/R&D—funding (regional) |

Source: Prepared by the Authors based on their own research of the Internet domains of businesses.

Table 6 shows the most important and disseminated CSR statements of ten Czech luxury fashion businesses via www statements e-published during Spring 2020, i.e., during the COVID-19 pandemic. They were researched on the domains of these business and based on the key words "name of business" and "COVID-19". Several businesses, including Blažek, Kara and Pietro Filipi, have completely omitted to issue any CSR statements and demonstrated a manifest lack of interest in the CSR, and this even with respect to marketing.

Interestingly, an overview of the 2018 and 2020 statements implies certain patterns and similarities allowing for discussing possible trends.

### 4.3. New Trends in CSR and CSR Reporting

The famous fashion designer Karl Lagerfeld even more famously stated, "Fendi is fur—and fur is Fendi . . . The reality is that they made fur so politically incorrect that, in the end, people do the opposite" [84]. This statement came in 2012/2013, that is, in the aftermath of the crises of 2007/2009. The fun fur concept by Fendi immediately clashed with the Gucci/Prada fur-free campaign (Fur Free Alliance, 2020a). Since then, there are two CSR categories highly important for international luxury fashion businesses—employees and environment, followed by the R&D category. This does not apply to Czech fashion businesses, which have demonstrated a rather weak (if any) commitment to CSR and CSR reporting. Then the COVID-19 pandemic and related crises came and fashion businesses needed to react—some of them passed on CSR and remained passive or worked more zealously on economic prospects. However, there were fashion businesses determined to demonstrate their commitment to the stakeholder model and to support financially, production-wise, or otherwise, the community, especially those fighting actively against

the COVID-19 pandemic. This sounds byzantine and fragmented, but the case study data with the holistic meta-analysis point to six new trends in CSR and CSR reporting.

First, all international luxury fashion businesses have been, and continue, paying particular attention to environmental and employee matters CSR categories and their statements have declaratory active features [4], while national Czech luxury fashion businesses are not really interested in these two and even not in other CSR categories, with the exception of Alpine Pro, Moira, and Timo. In full compliance with that, the concept of circular premium is much more developed on the international level than on the national level [33].

**Table 6.** Case Study—Ten Czech fashion businesses and their 2020 CSR statement.

| Group | CSR Statement | Comments |
|---|---|---|
| Alpine Pro | - | - |
| Bandi V. | - | - |
| Blažek P. | - | - |
| Evona | "The company Evona Chrudim decided to cease its production of all goods and instead make face masks to be delivered for free to the municipality . . . " | Social—very strong and altruistic |
| Kara | - | - |
| Moira | "Moira, . . . has delivered thousands of vestment sets to health workers using protection outfits . . . A part of face masks Moira was delivered for free to hospitals . . . The remaining face masks are offered via the e-shops. Currently, the face masks represent 50% of the entire productions. | Social—very strong and altruistic Economic—adjusting to the demand |
| Pietro Filipi | - | - |
| Timo | "Instead of bra, Timo in Litoměřice makes face masks for Prague. They will go e.g., to public transport drivers." | Social—very strong |
| Tonak | "For one cloth line face mask, the Novojičín municipality pays CZK 133 . . . At the very begin, when we desperately needed face masks, Tonak provided us with approx... 200 face masks for free . . . " | Social—but not altruistic Economic—adjusting to the demands |
| Triola | "Triola has a 100 years long tradition and now it goes for nano-industry. The turn was accelerated by the Coronavirus . . . And by the demand for health means . . . " | Triola |

Source: Prepared by the Authors based on their own research of the Internet domains of businesses.

Secondly, all international luxury fashion businesses made a CSR move in reaction to COVID-19, except Escada and Tod's. Highly interestingly, this move by the remaining eight international businesses was identical—providing donations and funding for medical and other research aimed at fighting against COVID-19. Pretty consistently, global businesses opted for the support of global projects while Italian businesses opted for the support of Italian projects.

Thirdly, half of the examined Czech businesses have not shown any signs of CSR reaction or even business conduct changes due to the COVID-19 pandemic. These five businesses have been consistently demonstrating a very reduced interest in CSR and CSR reporting. Indeed, Bandi Vamos, Blažek Praha, and Pietro Filipi all focus on formal fashion, especially male fashion, and appear to not be interested in sustainability issues even despite the COVID-19 pandemic.

Fourthly, the other half of the examined Czech businesses became very engaged due to the COVID-19 pandemic. Moira and Timo have organically accelerated their pre-existing commitment for CSR, while for Evona, Tonak, and Triola, the COVID-19 pandemic was rather a milestone, that is, a turning point. They all opted not for donations, but for action, that is, they started to make face-masks and other equipment needed during the COVID-19

pandemic. Unlike international luxury fashion businesses, they did not hesitate to start to produce items totally unrelated to the luxury fashion industry.

Fifthly, while international luxury fashion businesses integrated their (financial) help in their CSR program, that is, a donation for external medical research became one of many items on the payroll of their support programs, Czech businesses—if they decided to react to the COVID-19 pandemic—changed themselves and their production, not just temporarily, for example, Triola moved to nano-materials.

Sixthly, international luxury fashion businesses maintain an exclusivity picture and their CSR endeavors are rather in the format of the charity regarding somebody else (animals, poor people, sick people) while Czech businesses either basically skip the CSR or became actively engaged with their customers, that is, it is almost an "all or nothing at all" approach.

## 5. Aftermath Observations

The COVID-19 pandemic and its waves keep dramatically impacting the global society, and in particular the EU, even in the Spring of 2021. The luxury fashion industry keeps struggling and in the Czech Republic traditional fashion businesses, especially those without the CSR commitment, are directly threatened by the worsening situation.

Globally, since 2019, the interest in sustainability has reached and maintained an impressive level of 85%, namely 85% of investors from the general population consider the sustainable dimension of their, realized or planned, investments. Even more impressively, 95% of Millennials now express interest in sustainable investing [104]. From the economic and profit perspective, it is highly inspiring that 86% of investors believe that corporate Environment Social Governance ("ESG") practices can potentially lead to higher profitability and may be better long-term investments [104]. So far, it appears that this attitude is not crippled by the COVID-19 pandemic.

Regarding consumers, empirical studies regarding the fashion industry suggest that there is a recent clear trend regarding CSR awareness and readiness to pay extra, for example, as "the circular premium" [33], especially by those who are working in the industry [26]. Hence, the COVID-19 pandemic underlines the importance of CSR for the fashion industry, especially its luxury segment, and this from both the investor and consumer perspectives.

The message from the outside stakeholders is clear, but what about the fashion businesses themselves and their inside stakeholders, such as the management and employees? Fashion businesses clearly struggle and face a myriad of challenges and problems, including supply and demand disruptions and often a dramatic decline in the buying interest, especially in the high price and luxury segment [31]. This extremely threatening status receives much interest from the top management, but nets a rather weaker interest by the low management and some employees seem totally indifferent [6,7]. Indeed, the ignorance and underplaying of the CSR during the COVID-19 pandemic has become obvious, especially by certain Czech fashion businesses, see Tables 4 and 6 regarding Blažek, Kara, and Pietro Filipi.

Einstein's suggestion that "crisis brings progress" needs to be still appreciated in this context and read to its end, that is, that "in crisis, inventiveness, discovery and great strategy are born" [33]. Thus, *a contrario*, it can be suggested that, without inventiveness, originality, and perhaps CSR engagement as well, an industry, at least the fashion industry, might be doomed and be destroyed in the COVID-19 wave. Boldly, the lack of interest in sustainability, especially for the "reuse-recycle-recovery" circular concept, is not excusable because of COVID-19, instead, it is fatal during the COVID-19. This *prima facia* grim suggestion is already supported by some sad Czech evidence. Namely, Blažek, Kara, and Pietro Filipi businesses underplaying CSR and CSR statements, see Tables 4 and 6, and landing at the bottom of the Czech studied group in this respect have also become the first victims of the COVID-19 impact—they are facing bankruptcy, their clientele has vanished and their creditors seem to be determined to go ahead with their claims even if this could

lead to a total liquidation of these businesses [105–107]. Naturally, this observation might be incidental and not systemic, but it is definitely worth mentioning and using it to show that the collected CSR information matches exactly with the given proposition—ignoring CSR and CSR statements is definitely not a good idea during the COVID-19 pandemic and the fight merely for the economic pillar is not sustainable, that is, ignoring and "saving" on sustainability and CSR means a pathway to death for Czech luxury fashion businesses during the third wave of COVID-19 in the Spring of 2021.

In general, the performed field search and practical studies from 2018 to 2020, along with the 2021 aftermath observations suggest that the COVID-19 pandemic has changed the perception of CSR especially by international luxury fashion businesses which are inclined to a (more) sustainable approach to production and the sales process. Hence, we can suggest that COVID-19 has become a positive factor in the CSR development for International luxury fashion businesses. However, this is definitely not the case for Czech luxury fashion businesses and especially those most resistant might pay the highest price for that—end up in liquidation, *penny-wise and pound-foolish*.

## 6. Conclusions

The assessment of the of CSR statements of international and Czech Luxury Fashion Industry businesses (i) at the onset of the COVID-19 pandemic and (ii) during the COVID-19 pandemic has provided a very colorful and dynamically evolving picture underlying the differences in the perception and recognition of the importance of the CSR and CSR statements by International and Czech luxury fashion businesses. The afore-mentioned much stronger CSR statements by the international businesses as opposed to Czech businesses at the onset of the COVID-19 pandemic have been magnified by the observed wording of statements during the COVID-19 pandemic in 2020 and by the aftermath observations from 2021.

Further, this allowed for the identification of six new trends. In sum, international luxury fashion businesses seem to homogenously go more and more for extrinsic CSR which does not really influence their exclusive and lavish line of production and can impact only the materials and distribution channels (no fur, less defilés de mode) and COVID-19 confirmed this pattern. Czech fashion businesses attempting to compete with these international luxury fashion businesses have a rather heterogenous approach to CSR and their differences became magnified by COVID-19. Some of them totally ignored CSR, while others embraced it more than vigorously and made it truly intrinsic, so even this could be perceived as a potential challenge for the exclusivity and lavish aura.

Arguably, the three stages of the CSR evolution can be perfectly demonstrated by the performed case study [38]—the CSR cultural reluctance (Czech formal—Bandi Vamos, Blažek Praha, etc.), the CSR cultural grasp (International luxury fashion businesses), and the CSR cultural embedment (Czech underwear—Evona, Moira, Timo, Triola, etc.). This leads to propositions that COVID-19 has been perceived (i) either as a threat to be passively survived by the Czech men's formal fashion businesses, or (ii) an opportunity to give another touch, an aura, of ethics to international luxury fashion businesses and slow down a little bit more (iii) or call for the Czech women's fancy underwear businesses, which truly took on COVID-19 as an impulse for rebuilding their production portfolio. These three options are far from perfect. The first option is regretfully passive and rigid. The second option lags behind expectations and misses the multi-stakeholder model. The third option is perhaps too revolutionary for the conservative, exclusive and lavish fashion setting, that is, can fire back.

Naturally, the performed case study extended to only 20 luxury fashion businesses and their CSR statements in 2018 and 2020. This limitation could be offset by expanding the sample (larger study), expanding the observed period (longitude study), and by checking not only words but actions as well (behavior study). Nevertheless, the extension of the sample would mean the loss of its homogeneity and would inevitably lead to the departure from this particular focus while preferring the comparison. Therefore, this

limitation perhaps is inherently inevitable for this type of study, and this in particular due to the concept of luxury fashion uniqueness [14,29,38,66]. In contrast, the longitude and behavioral features could and should be rather smoothly included and addressed as appropriate in future studies.

Despite these limitations, the nature of the sample and consistency of the results make the proposed six trends worthy of further exploration and the proposed three options worthy of several propositions—to enhance awareness, to engage deeper in the multi-stakeholder model, and to check whether the *étatique*, customer and business vision, mission, and expectations match. Definitely, the time has arrived to slow down the fast fashion business and to find a new balance—to reconcile the CSR commands for the 21st century and to make the luxury mode a truly slow mode lavishly *Nachhaltig.*

**Author Contributions:** Conceptualization, R.M.P., T.N. and R.K.M.; methodology, R.M.P.; software, R.M.P.; validation, R.M.P. and R.K.M.; formal analysis, R.M.P. and T.N.; investigation, R.M.P. and R.K.M.; resources, R.M.P. and T.N.; data curation, R.M.P.; writing—original draft preparation, R.M.P.; writing—review and editing, R.M.P. and T.N.; visualization, R.M.P.; supervision, R.M.P. and R.K.M.; project administration, R.M.P.; funding acquisition, R.M.P. All authors have read and agreed to the published version of the manuscript.

**Funding:** This research and resulting paper are the outcome of Metropolitan University Prague research project no. 87-02 "International Business, Financial Management and Tourism" (2021) based on a grant from the Institutional Fund for the Long-term Strategic Development of Research Organizations.

**Institutional Review Board Statement:** Not applicable.

**Informed Consent Statement:** Not applicable.

**Data Availability Statement:** Data available in a publicly accessible repository that does not issue DOIs. Publicly available datasets were analyzed in this study. This data can be found here: https://justice.cz/ & https://e-justice.europa.eu/ (accessed on 3 March 2021).

**Acknowledgments:** The authors are grateful for the ongoing institutional support arranged by the Centre for Research Support at the Metropolitan University Prague, especially Tereza Vogeltanzová and Hana Raková, and highly relevant useful comments and suggestions provided during the peer-review.

**Conflicts of Interest:** The authors declare no conflict of interest.

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
