# Peer review of "CSR Statements in International and Czech Luxury Fashion Industry at the Onset and during the COVID-19 Pandemic—Slowing Down the Fast Fashion Business?"

_sustainability, doi:10.3390/su13073715_

Round 1

Reviewer 1 Report

The subject of the research undertaken is very interesting, especially because of the very current economic crisis caused by COVID-19.

The goals of the peer-reviewed article were clearly defined.

Linking the world of fashion with sustainable development seems to be difficult to do, but the authors of the peer-reviewed article showed that such a link is possible and in a very skilful way showed sustainability based on three pillars, i.e. economic, environmental and social.

The weak point of the presented analyzes is the low size of the research sample, but it is difficult to object to the authors on this account, because it must be remembered that fashion companies represent rather a corporate model and this means their significant spread and creating a unique brand, which limits the number of companies operating on the market from the fashion industry.

Author Response

Reply to Reviewers

Manuscript ID: sustainability-1152633

Title: CSR Statements in International and Czech Luxury Fashion Industry at the onset and during the COVID-19 pandemic – Slowing Down the Fast Fashion Business?

Authors: Radka MacGregor Pelikánová *, Tereza Němečková, Robert Kenyon MacGregor

We are very grateful for the provided peer-review and valuable recommendations which we have gladly reflected and addressed as follows (our changes in the manuscript are highlighted in yellow):

Sustainability Peer Review 1

Thank you very much for your nice and encouraging words, we appreciate that you share our opinion about the dynamic potential for an interaction between the CSR and the COVID-19 impact.

Yes, you are absolutely right that the weak point of our research, study and manuscript is the small size of the research sample. However, and exactly as you added, this weak point is inherent, because the fashion industry, and in particular the luxury fashion industry, does not entail a large number of businesses or to put it differently, a large number of businesses labelled as the luxury fashion businesses would mean a contradiction in terms and the rejection of the brand uniqueness. Nevertheless, this very relevant point presented by the Reviewer 1 has motivated us to address this issue in the “3. Materials and Methods” and in “6. Conclusions” in more depth and while adding references.

Sustainability Peer Review 2

Thanks for sharing with us your point of view about the importance of the CSR monitoring during the COVID-19 pandemic.

Your concern about the selection and size of the sample is similar to the concern expressed by Reviewer 1 and we have addressed in “3. Materials and Methods” and in “6. Conclusions” in more depth and while adding references.

Following your advice, we have provided additional information about the performance of the qualitative research and assessment by the selected three experts – different from the Authors – via the simplified Delphi method and Meta-Analysis. This was completed in “3.Materials and Methods”.

Sustainability Peer Review 3

Thank you very much for your compliments and encouraging words, we are glad that you appreciate our efforts.

Yes, we fully agree with you regarding your 1st recommendation, the circular premium deserves to be mentioned even in the discussion and we have incorporated a short comment in “4.3. New Trends in CSR and CSR reporting”.

We still believe that the section 5 should carry the title “Aftermath Observations” while section 4 should carry the title “Results and Discussion.”

Your concern about the limit(ation) of our research and study has been mirrored as well by Reviewer 1 and we do recognize that this deserves more explanation. Consequently, we have updated “6.Conclusions” and developed the information about the uniqueness and implied limitations 3. Materials and Methods” and in “6. Conclusions” in more depth and while adding references.

Thanks again for your feedback,

RMP, TN and RKM

Reviewer 2 Report

Currently, the impact of Covid-19 topic is significant in all areas of the economy. Its impact on CSR is very important to monitor, as it can have a long-term impact on the sustainability of companies, industries sectors and societies. In individual countries where the production of luxury goods is located, there is a significant impact on the environmental and social sphere of CSR. In countries where luxury brands are sold, it is important to monitor reporting as stakeholders are interested. The topic and focus of the article is therefore important because it compares the same number of companies that only sell luxury goods in the Czech Republic and those that are originally from the Czech Republic.

However, it would be appropriate to highlight and describe how these companies were selected or a closer specification of the orientation to export or not and possible their basic indicators of the financial situation from the financial statements (for example, sales or profit) would help. The possible effects of this information on the evaluation may be interesting.

The methodology also requires a more description of whether the qualitative research was carried out simultaneously by all authors on all companies and the subsequent joint unification, or whether the research was divided and how, describe a way of results unification.

Author Response

(The authors gave the same response as above.)

Reviewer 3 Report

Dear authors

this work is really well elaborated. My congratulations. The concept is well proposed, the novelty of the paper is clear, the methodology used is explained in all details, it is showed a critical analysis of results, there is a discussion of several points.

However I suggest only three little changes:

  1. The concept of circular premium should be proposed and investigated in this work (also an issue to enhance)
  2. The name of section 5 Discussions
  3. The absence of limits of the work in the last section

Author Response

(The authors gave the same response as above.)
